# Patellar Osteomyelitis in a 9-Year-Old Patient with Chronic Granulomatous Disease: A Case Report

**DOI:** 10.3390/children9010076

**Published:** 2022-01-05

**Authors:** Yonggeun Park, Seungjin Yoo, Yongyeon Chu, Chaemoon Lim

**Affiliations:** Department of Orthopedic Surgery, Jeju National University Hospital, Jeju 63241, Korea; pyk184@hanmail.net (Y.P.); syoo06@gmail.com (S.Y.); yy-chu@hanmail.net (Y.C.)

**Keywords:** patellar osteomyelitis, chronic granulomatous disease, immunodeficiency disorder

## Abstract

Hematogenous osteomyelitis is commonly reported in long tubular bones in the pediatric population. Acute osteomyelitis involving the patella is extremely uncommon in children, and its diagnosis is frequently delayed due to its rarity and variable clinical manifestations. Chronic granulomatous disease (CGD) is a rare genetic immunodeficiency disorder characterized by severe recurrent bacterial and fungal infections. The most commonly affected sites of infection are the lungs, lymph nodes, skin, liver, and gastrointestinal tract. Acute hematogenous osteomyelitis of the patella associated with CGD has never been reported. Our report describes the first case of acute hematogenous patellar osteomyelitis in a pediatric patient with CGD. Her clinical manifestations were similar to other possible differentials such as septic arthritis; however, use of advanced imaging confirmed the diagnosis, and the patient was successfully managed surgically. Since hematogenous osteomyelitis in children is uncommon, a high index of suspicion and advanced imaging may help with its diagnosis, and in cases where antibiotic treatment proves to be insufficient, prompt surgical management is imperative.

## 1. Introduction

Hematogenous osteomyelitis is commonly reported in long tubular bones and occasionally in cuboidal, vertebral, and pelvic bones in the pediatric population [1,2]. However, acute patellar osteomyelitis is extremely uncommon in children, and its diagnosis is frequently delayed due to its rarity and variable clinical manifestations in children [3]. Consequently, early diagnosis of acute patellar osteomyelitis is the key to treatment success using appropriate antibiotics and occasionally surgical intervention in order to prevent debilitating complications involving bone or joint destruction [2].

CGD is a rare genetic condition of congenital immunodeficiency in which an underlying defect of nicotinamide adenine dinucleotide phosphate oxidase complex fails to generate reactive oxygen species in activated phagocytes such as neutrophils, monocytes and macrophages. Patients with CGD are unable to eradicate certain catalase-positive microorganisms including bacteria and fungi, resulting in recurrent infection [4]. The most commonly affected sites of infection are the lungs, lymph nodes, skin, liver, and gastrointestinal tract. Osteomyelitis is a rare complication of CGD, and most incidences of osteomyelitis occur in the ribs and vertebrae secondary to contiguous spread from lung abscess [5,6]. *Aspergillus* spp. is reportedly the most common organism associated with osteomyelitis in CGD, and skeletal infection in CGD is frequently associated with pulmonary aspergillosis, although the exact mechanism is unknown [6].

Herein, we report the first case of acute patellar osteomyelitis in a 9-year-old girl with CGD, who was successfully managed surgically. To our knowledge, the current case is the first report on primary patellar osteomyelitis without an adjacent infection focus in a child with CGD. Her clinical presentation and intraoperative findings are explained and discussed in detail, in addition to a review of patellar osteomyelitis in the pediatric population.

## 2. Case Report

A 9-year-old girl with CGD was referred to our department from the department of pediatrics with a chief complaint of difficulty in ambulation due to left knee pain persisting for two days without antecedent trauma. She had been hospitalized for recurrent cervical lymphadenitis and multiple liver abscesses and had been on intravenous teicoplanin and cefotaxime. Physical examination revealed painful limitation of motion of the left knee aggravated by full extension, swelling, heating sensation, and tenderness with no penetrating skin injuries. Initial laboratory findings are shown in Table 1. All attempts at joint aspiration including sonography-guided aspiration in the left knee failed.

A Simple radiograph of the left knee showed no definite bony abnormalities other than moderate soft tissue swelling (Figure 1). Subsequent gadolinium-enhanced magnetic resonance imaging (MRI) revealed peripheral enhancement of subcutaneous fat at the prepatellar area, bone marrow edema with enhancement at the patella, and a small amount of joint effusion with diffuse synovitis in the left knee joint (Figure 2).

With the initial impression of infectious prepatellar bursitis, patellar osteomyelitis, and septic arthritis of the left knee, arthroscopic debridement and open incision and drainage were recommended; however, her parents strongly refused and opted for conservative management due to the patient’s immunocompromised status. Four weeks after the onset of knee pain, skin breakage with concurrent serous discharge appeared in the prepatellar area (Figure 3). Consequently, conservative management with antibiotics, wound management, and physical therapy was continued after discharge from hospital, but left knee pain, limitation in range of motion, and the skin lesion with persistent discharge remained.

One year and six months after the onset of symptoms, the parents and patient finally consented to surgical intervention for left patellar osteomyelitis due to persistent pain, infection, and functional deficit. On admission to the department of orthopedic surgery, initial laboratory findings are shown in Table 1. Physical examination revealed mild swelling on the left knee along with a slightly improved skin lesion and persistent discharge (Figure 4). Simple radiograph of the left knee revealed irregular sclerotic change in the posterior cortex of the patellar bone (Figure 5), and gadolinium-enhanced MRI indicated heterogeneous enhancement of the patella with a fistula between the patella medullary canal and subcutaneous fat in the left patellar area (Figure 6).

Following diagnosis of chronic osteomyelitis and abscess in the left patella, open lavage and debridement of abscess and sequestrum at the patella were planned by an anterior approach without arthrotomy. Intraoperatively, active pus and hypertrophied synovium were noted upon skin incision, and swab culture was performed. Necrotic bone debris and sequestrum at the patella were thoroughly curetted and debrided with massive irrigation through medial and lateral bone defects on the anterior cortex of the patella along with tissue culture and bone biopsy (Figure 7). The intraoperative histologic examination confirmed chronic osteomyelitis, but no microorganism was cultured on both swab and tissue specimens. Postoperatively, the patient was treated with intravenous cefotaxime for four weeks and discharged without further use of oral antibiotics upon consultation with the pediatric infectious disease specialist.

Two weeks after the discharge, follow-up laboratory investigations conducted at the outpatient clinic revealed WBC, CRP, and ESR within normal ranges, and the clinical examination was unremarkable without pain (Table 1). At the latest follow-up on the postoperative 15th month, the patient remained symptom-free with a full range of knee joint motion, and simple radiograph showed nearly full bone recovery of the patella (Figure 8).

## 3. Discussion

In this study, we reported an extremely rare case of osteomyelitis with patellar involvement in a 9-year-old patient with CGD, a congenital immunodeficiency disorder. Surgical treatment of chronic patellar osteomyelitis was significantly delayed by the parents’ refusal due to the patient’s underlying immunocompromised status, invasive bacterial infection involving multiple organs, and recurrent liver abscess. To our knowledge, no incidence of patellar osteomyelitis has been reported in pediatric patients with CGD.

Aside from its uncommon incidence rate of 2.9 per 100,000 children, the age of peak incidence of osteomyelitis is reportedly between 5 and 15 years [1,7,8]. The patellar bone makes a radiographic appearance between 4 and 6 years, and the ossification of its growth cartilage is complete around 12 years. It also has a rich arterial anastomotic network comprising of the superior and inferior genicular arteries and the anterior tibial recurrent arteries. This vast extraosseous arterial network begins to appear around age 5 and reaches its maximal anastomosis by 12 years [1]. In this age group between 5 and 12 years, patellar osteomyelitis without occult trauma is most commonly caused by the hematogenous spread of infection. However, the rarity of this condition in pediatric population can be explained by the high number of immune cells in its rich vascular anastomotic complex, its characteristic hemodynamics lacking formation of vascular loops, which causes bacterial trapping, the absence of physeal plate, and a limited space for bacterial growth within the medullary canal [1,7,8,9].

Accurate diagnosis of patellar osteomyelitis is challenging and often delayed due to its rarity and variable clinical manifestations. Initially suspected diagnoses usually include septic arthritis, cellulitis, bursitis, and synovitis [1]. Children with patellar osteomyelitis present with variable degrees of subjective pain, limitation of joint motion, and associated gait disturbance. In addition, initial laboratory and simple radiographic findings are generally unremarkable and do not point to the diagnosis of patellar osteomyelitis [1,7,10]. Misdiagnosis and delayed diagnosis of this condition result in its progression to chronic osteomyelitis and subsequent septic arthritis. Therefore, a high index of suspicion, close clinical observation, and advanced imaging modalities such as bone scintigraphy, computed tomography, and gadolinium-enhanced MRI are highly recommended for prompt and accurate diagnosis of patellar osteomyelitis [8,9]. In this case, despite an initial suspicion of septic arthritis of the knee joint, patellar osteomyelitis was eventually diagnosed using gadolinium-enhanced MRI.

Currently, treatment guidelines or recommendations for patellar osteomyelitis are not well established. Most cases of hematogenous osteomyelitis in pediatric patients are managed using antibiotics, but the presence of abscess or sequestrum requires surgical debridement [3]. Our review of available literature revealed that previous reports on patellar osteomyelitis in the pediatric population were mostly limited to case reports and small case series. We found 18 previous studies, comprising 28 pediatric patients diagnosed with patellar osteomyelitis, and detailed information on their demographics, treatment, and culture results is summarized in Table 2.

In the literature, although few cases were confirmed despite the absence of bacterial growth, the most commonly cultured pathogenic organism was *Staphylococcus aureus*. In addition, unlike other forms of hematogenous osteomyelitis, 22 of the 28 (78.6%) patients underwent surgical management for hematogenous patellar osteomyelitis. Its rarity and difficulty in diagnosis, delay in commencement of antibiotics, or subsequent fistula or abscess formation may have contributed to the relatively higher rate of surgical management [9]. Surgical treatment with debridement and curettage is recommended in patients with failed antibiotic treatment and in patients with complications, as seen in the present case [3]. In addition, the application of a chain of antibiotic impregnated beads was reported with favorable outcomes in one case of delayed diagnosis of patellar osteomyelitis; however, the disadvantage of this method requires a second operation for beads removal [2].

## 4. Conclusions

In conclusion, we report an extremely rare case of hematogenous osteomyelitis with patellar involvement in a 9-year-old girl with a congenital immunodeficiency disorder. Despite frequent reports on rarity, delay, and difficulty in diagnosis of patellar osteomyelitis, a high index of suspicion, close clinical monitoring, and application of advanced imaging modalities such as gadolinium-enhanced MRI are highly recommended. Although antibiotic treatment is generally sufficient for the management of osteomyelitis in pediatric patients, prompt surgical treatment is reserved for those with delay in diagnosis and persistence in spite of antibiotics treatment in pediatric patellar osteomyelitis.

## Figures and Tables

**Figure 1 children-09-00076-f001:**
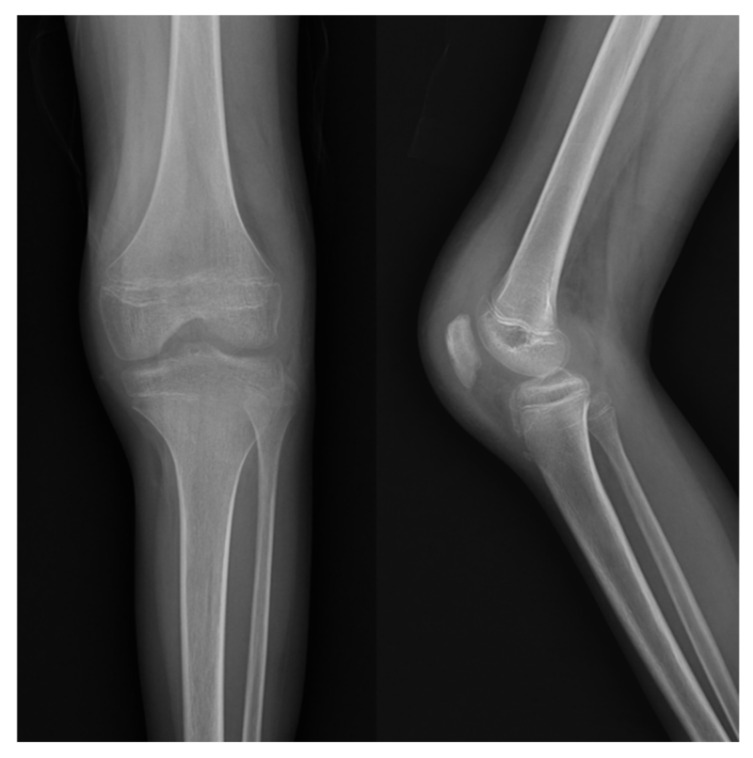
Initial simple radiograph of the left knee revealed no definite bony abnormalities other than moderate soft tissue swelling.

**Figure 2 children-09-00076-f002:**
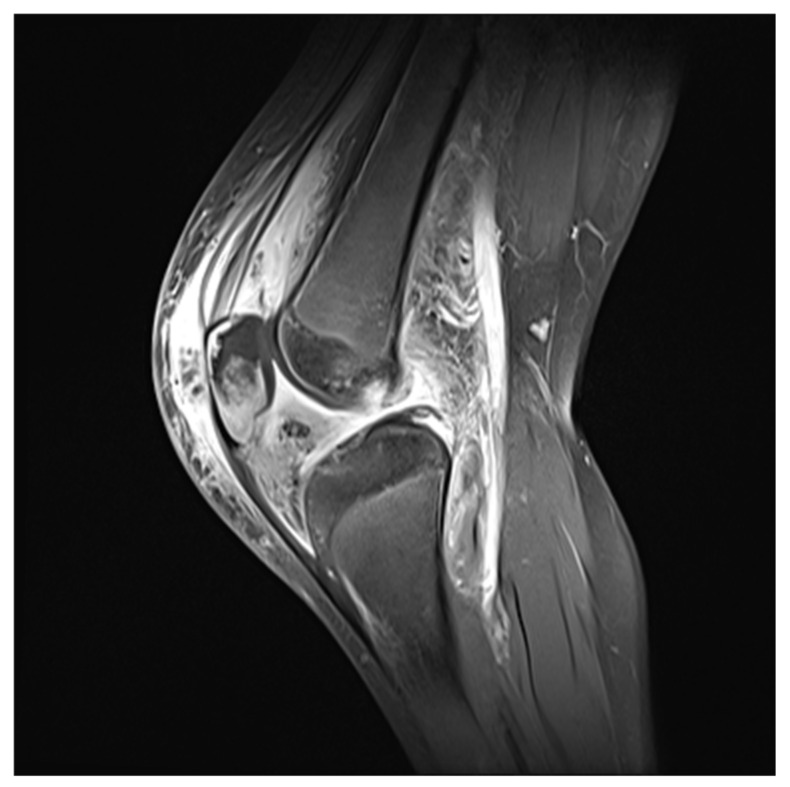
Initial gadolinium-enhanced MRI showed peripheral enhancement of subcutaneous fat at the prepatellar area, bone marrow edema with enhancement at the patella, and a small amount of joint effusion with diffuse synovitis in the left knee joint.

**Figure 3 children-09-00076-f003:**
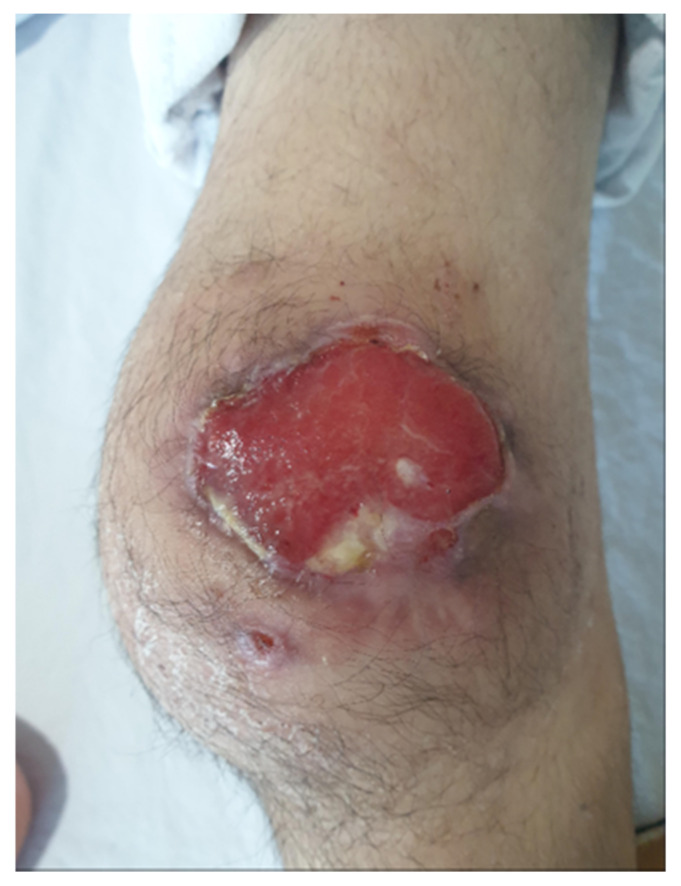
Skin breakage with concurrent serous discharge appeared in the prepatellar area 4 weeks after the onset of initial symptom.

**Figure 4 children-09-00076-f004:**
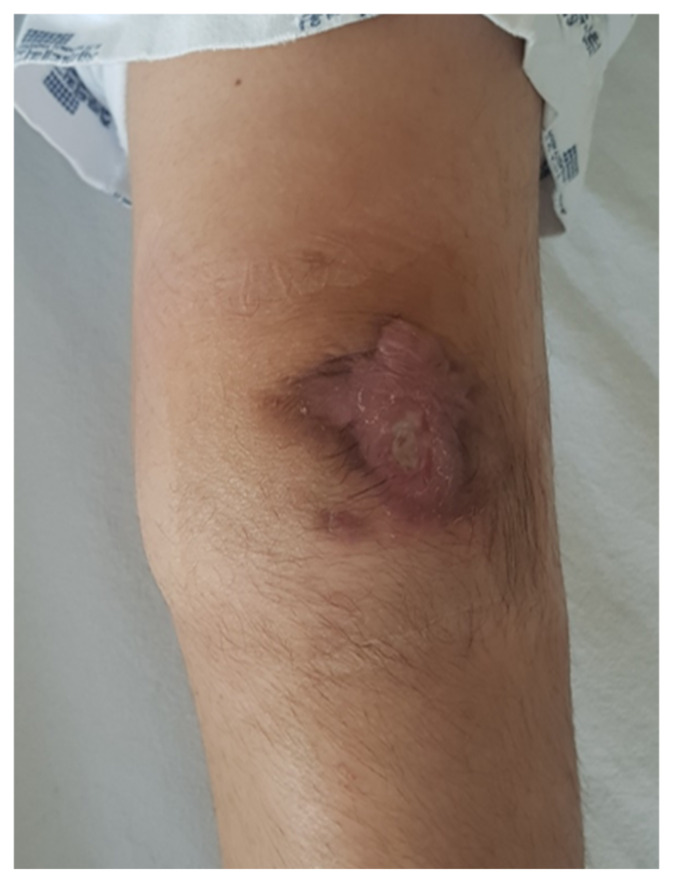
One year and six months after the onset of symptoms, gross examination of the knee revealed mild swelling on the left knee along with a slightly improved skin lesion and persistent discharge.

**Figure 5 children-09-00076-f005:**
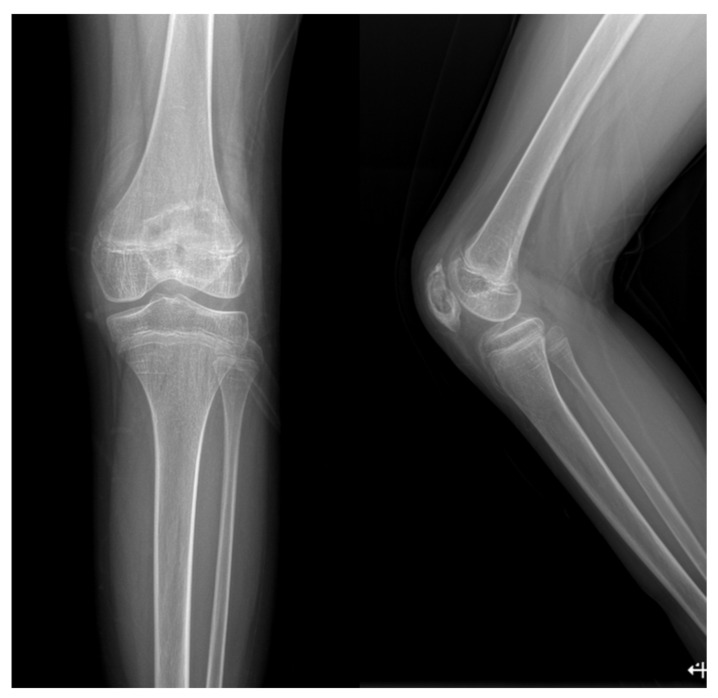
Preoperative simple radiograph of the left knee showed irregular sclerotic change in the posterior cortex of the patellar bone.

**Figure 6 children-09-00076-f006:**
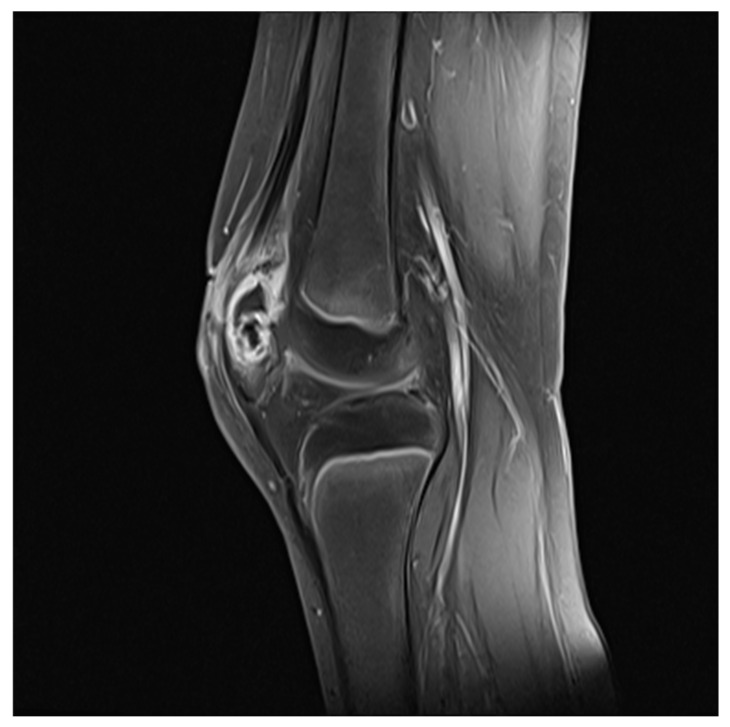
Preoperative gadolinium-enhanced MRI indicated heterogeneous enhancement of the patella with a fistula between the patella medullary canal and subcutaneous fat in the left patellar area.

**Figure 7 children-09-00076-f007:**
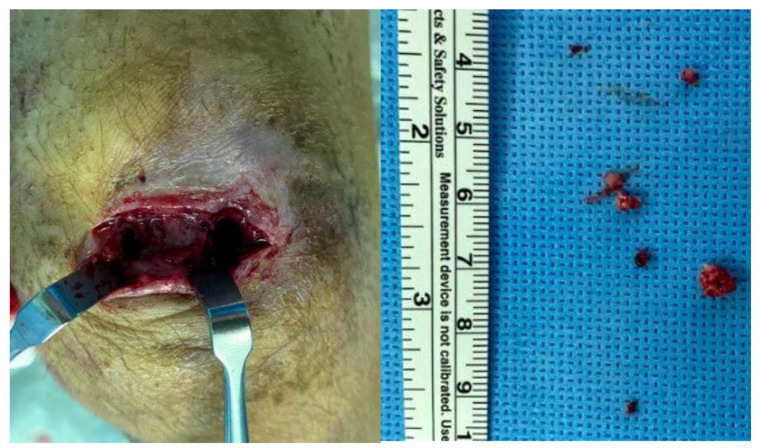
Intraoperative findings showed necrotic bone debris and sequestrum at the patella through medial- and lateral-sided bone defects on the anterior cortex of the patella.

**Figure 8 children-09-00076-f008:**
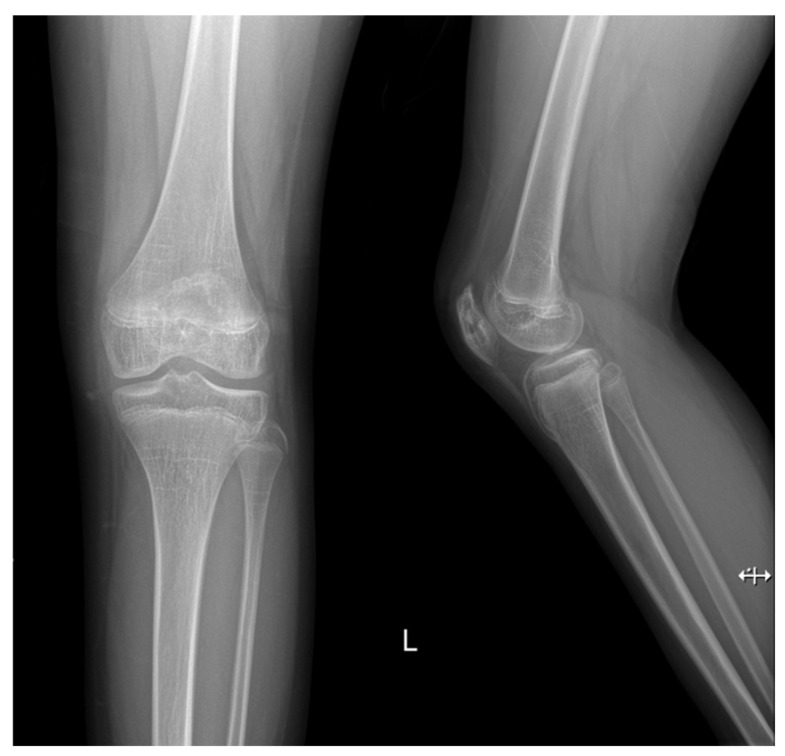
Simple radiograph of the left knee performed at the postoperative 15th month indicated bony recovery of the patella.

**Table 1 children-09-00076-t001:** Summary of clinical evolution and laboratory findings in a chronological order.

Date	White Blood Cell Counts (/µL)	Neutrophils (%)	C-Reactive Protein (mg/dL)	Erythrocyte Sedimentation Rate (mm/h)
2018-10	10,000	85.6	10.14	120
	Initial presentation of knee pain and difficulty in ambulation			
2018-11	10,300	52.4	5.03	120
	4 weeks after the use of teicoplanin and cefotaxime since initial presentation			
2020-05	7100	54.6	0.37	19
	Admission to the department of orthopedic surgery before surgery			
2020-06	7200	46.9	0.17	26
	Postoperative 4th week after use of cefotaxime for 4 weeks			
2020-06	7500	55.6	0.35	39
	Postoperative 6th week at out-patient clinic follow-up after discharge			
2021-06	8000	58.5	0.34	29
	Postoperative 1st year at out-patient clinic follow-up			

**Table 2 children-09-00076-t002:** Literature review for patellar osteomyelitis in children.

Author	Cases		Operative Treatment	Culture
	Age	Sex		
Evans 1962 [11]	5	F	No	No growth
	7	F	No	*Staphylococcus aureus*
	7	M	Yes	*Staphylococcus aureus*
Angella 1967 [10]	7	F	Yes	*Staphylococcus aureus*
	11	M	No	No growth
Wadlington et al. 1971 [12]	9	M	No	*Pseudomonas aeruginosa*
Vaninbroukx et al. 1976 [13]	3	F	Yes	Not available
	6	M	Yes	No growth
	5	M	Yes	Not available
Cahill 1978 [8]	7	M	Yes	Gram positive cocci
	8	M	Yes	No growth
Papavasiliou et al. 1989 [14]	6	F	Yes	*Staphylococcus aureus*
	7	M	Yes	*Staphylococcus aureus*
	8	M	Yes	*Staphylococcus aureus*
Roy et al. 1991 [1]	9	F	Yes	*Staphylococcus aureus*
	6	F	Yes	*Staphylococcus aureus*
	7	M	Yes	*Staphylococcus aureus*
	8	M	Yes	*Clostridium bifermentans*
Moyikoua et al. 1993 [15]	8	M	No	Not available
Masuda et al. 1999 [16]	8	M	Yes	No growth
Durani et al. 2006 [17]	7	F	Yes	*Staphylococcus aureus*
Sperl et al. 2007 [2]	10	F	Yes	*Staphylococcus aureus*
Choi et al. 2007 [7]	9	M	Yes	No growth
	7	M	Yes	*Staphylococcus aureus*
De Gheldere 2009 [18]	10	M	Yes	*Staphylococcus aureus*
Gil-Albarova et al. 2012 [19]	8	F	Yes	*Staphylococcus aureus*
Tanikawa et al. 2017 [9]	9	F	Yes	*Staphylococcus aureus*
Traverso et al. 2020 [20]	12	M	No	*Staphylococcus aureus*

## Data Availability

The data presented in the case report are available on request from the corresponding author.

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
