# Peer review of "Patellar Osteomyelitis in a 9-Year-Old Patient with Chronic Granulomatous Disease: A Case Report"

_children, 2022, doi:10.3390/children9010076_

Round 1

Reviewer 1 Report

The case is interesting and it is worth reporting, but the section "case report" is too long.

In my opinion, the discussion must be focused on patellar osteomyelitis, a rare event in pediatric patients. All is reported between line 179 and 189 is redundant and should be summed up. 

Author Response

We have revised the “case report” section by removing unnecessary details on clinical evolution during preoperative evaluation and by making the case report section more concise. In addition, we have removed the redundant part on chronic granulomatous disease in the discussion section (between line 179 and 189) in order to solely focus on the patellar osteomyelitis in pediatric population.

Reviewer 2 Report

Dear Authors,

This is a very interesting case report. The authors aimed to present the first case of acute hematogenous patellar osteomyelitis in a pediatric patient with chronic granulomatous disease. The case is well presented. The discussion is interesting and well written.

Nevertheless, I have some suggestions to improve the paper:

  1. I suggest considering adding the table with the laboratory tests results.
  2. The English language should be corrected because there are some spelling and grammar mistakes that make the article

Author Response

We have formulated a table for laboratory findings in a choronological order with a brief summary of clinical evolution (Table 1). Furthermore, we have improved the English language and proficiency of the manuscript with a review by a native speaker.